# Multimodality Imaging of Primary Hepatic Lymphoma: A Case Report and a Literature Review

**DOI:** 10.3390/diagnostics14030306

**Published:** 2024-01-31

**Authors:** Ryosuke Taiji, Nagaaki Marugami, Aki Marugami, Takahiro Itoh, Sho Shimizu, Ryota Nakano, Yuki Hoda, Hideki Kunichika, Tetsuya Tachiiri, Kiyoyuki Minamiguchi, Satoshi Yamauchi, Toshihiro Tanaka

**Affiliations:** Department of Diagnostic and Interventional Radiology, Nara Medical University, 840 Shijo-cho, Kashihara-city, Nara 634-8522, Japan; marugami@naramed-u.ac.jp (N.M.); aki08052@naramed-u.ac.jp (A.M.); titoh@naramed-u.ac.jp (T.I.); sho_shimizu.rad@naramed-u.ac.jp (S.S.); k106328@naramed-u.ac.jp (R.N.); yuki.hoda@naramed-u.ac.jp (Y.H.); k102972@naramed-u.ac.jp (H.K.); k135334@naramed-u.ac.jp (T.T.); kiyo829@naramed-u.ac.jp (K.M.); syamauchi@naramed-u.ac.jp (S.Y.); totanaka@naramed-u.ac.jp (T.T.)

**Keywords:** lymphoma, primary hepatic lymphoma, ultrasonography, contrast-enhanced ultrasonography, computed tomography, magnetic resonance imaging

## Abstract

Primary hepatic lymphoma (PHL) is a rare form of non-Hodgkin lymphoma primarily affecting the liver. We present a case of an 84-year-old man diagnosed with PHL, incidentally detected during abdominal ultrasonography. The ultrasonography showed a hypoechoic nodule. When examined by CEUS, the nodule showed hyperenhancement in the arterial phase and hypoenhancement in the portal and late phases. Conversely, CECT demonstrated hypoenhancement through all the phases. The patient declined a tumor biopsy and opted for follow-up care. Ten months later, the lobular mass had increased from 15 mm to 65 mm, presenting as hypoechogenic and demonstrating the “vessel-penetrating sign” on color Doppler imaging. CEUS revealed reticulated enhancement, indicating intratumoral vessels. The mass displayed hypoattenuation on plain CT, hypointensity in T1-weighted images, and hyperintensity in T2-weighted images and exhibited significant restriction in diffusion-weighted images. Both CECT and contrast-enhanced MRI exhibited hypoenhancement. The patient underwent a partial hepatic segmentectomy, and the mass was pathologically diagnosed as a diffuse large B-cell lymphoma. Subsequent postoperative radiological examinations revealed no other lesions, confirming the diagnosis of PHL. Our report highlights specific ultrasonographic signs of PHL observed from an early stage and presents a review of the relevant literature.

## 1. Introduction

Primary hepatic lymphoma (PHL) represents a rare malignancy, characterized as an extranodal lymphoma primarily affecting the liver and lacking lymphomatous infiltration into other organs, such as lymph nodes, bone marrow, or spleen. Its occurrence is reported to be less than 1% among extranodal malignant lymphoma [1]. Typically observed in individuals aged 50 to 60 years, PHL exhibits a higher prevalence among males than among females (with a ratio from 1.7 to 1.0) [2,3]. The clinical manifestations of PHL often include nonspecific symptoms such as nausea, fever, upper abdominal pain, or discomfort. Macroscopically, PHL can present in three main types: as a solitary mass, as multiple nodules, and as diffuse lesions without nodule/mass formations. Among them, the solitary mass type is the most commonly encountered [4].

Ultrasonography serves as the first examination for the screening of focal liver lesions [5]. Detecting a focal liver lesion requires further diagnostic techniques like color Doppler imaging, pulsed-wave Doppler imaging, and contrast-enhanced ultrasonography (CEUS). Color Doppler imaging assesses the vascular abundance in a mass, and pulsed-wave Doppler imaging analyzes the blood flow within the hepatic artery and portal vein [6,7]. CEUS aids in differentiating solitary liver masses. Reports on B-mode ultrasonographic findings in PHL generally depict hypoechoic lesions compared to the normal liver or reveal penetrating vessels [8,9]. Penetrating vessels can be visualized via color Doppler imaging or CEUS within a large PHL.

Unlike extracellular contrast agents used in contrast-enhanced CT (CECT) or MRI, ultrasound contrast agents circulate solely within blood vessels, allowing a more reliable view of intratumoral vascular structures [10]. In breast cancer, CEUS demonstrates higher sensitivity than MRI for evaluating microvessel density, including identifying penetrating vessels [11]. However, few PHL cases assessed with CEUS have been reported [12,13,14]. This report presents a PHL case with specific ultrasonographic findings like enhancement patterns and vessel-penetrating signs observed via CEUS from an early stage. It further reviews the literature encompassing the clinical, radiological, and pathological features of this tumor. To our knowledge, this may be the first report detailing enhancement patterns in the early stage of PHL on CEUS.

## 2. Case Presentation

An 84-year-old man underwent an abdominal ultrasonography scan for diabetes mellitus screening, which incidentally identified a hypoechogenic nodule in segment 7 of the liver. The patient did not complain of any symptoms, and his medical history included diabetes mellitus and hypertension. Notably, no history of alcohol consumption or blood transfusions was reported, and there were no remarkable medical records within his family history. On a blood test, the blood count was normal, biochemistry showed no elevation in liver function or cholestasis-indicating enzymes, and HbA1c was elevated at 7.3% (Table 1). Viral markers were negative. The levels of serum alpha-fetoprotein (AFP) and des-gamma-carboxy prothrombin (DCP) were within the normal limits among the tumor markers, while that of soluble interleukin-2 receptor (sIL-2R) was mildly elevated, being 599 U/mL.

Initial ultrasonography showed a well-defined lobular nodule 15 mm in size with low echogenicity, similar to the vessel in segment 7 of the liver, and color Doppler imaging depicted a linear blood flow within the nodule (Figure 1). A strand with intermediate echogenicity penetrating within the nodule was seen. Low-mechanical-index CEUS (low-MI CEUS) showed reticular vessels and gradual enhancement in the arterial phase, lower enhancement than in the surrounding liver parenchyma in the portal phase, and a complete lack of enhancement in the late phase (Figure 2). Plain CT demonstrated a hypointense nodule, and CECT depicted a homogeneous hypovascular nodule in both the arterial and the equilibrium phases (Figure 3).

The patient was recommended to undergo a liver biopsy, but he declined. Subsequently, he was followed up without any intervention. Ten months later, CT revealed a lobular heterogeneous mass of approximately 7 cm in size. Its peripheral region displayed an intermediate attenuation close to the liver parenchyma, while the central region showed hypoattenuation. In CECT images, the mass appeared as a hypovascular tumor, and the hepatic arteries in the posterior area penetrated the mass.

MRI also showed a heterogenous mass with hyperintensity in the central region and intermediate intensity in the peripheral area on T2-weighted images (T2WI) (Figure 4). Diffusion-weighted images (DWI) revealed a markedly high signal intensity with a low apparent diffusion coefficient (ADC) value of 0.5 × 10^−3^ mm^2^/s. There was no macroscopic adipose tissue, and chemical shift imaging did not reveal any fatty component in the mass. Gd-EOB-DTPA-enhanced MRI (EOB-MRI) displayed the hypovascular mass and no contrast agent uptake within the mass in the hepatobiliary phase (HBP).

The second ultrasonography 10 months after the initial ultrasonography, which was taken at the same time as the CT/MRI, showed a mass of 6.5 cm in size. The peripheral area of the mass was hypoechoic, comparable to the blood vessels, and the center of the mass showed a high echogenic area on B-mode ultrasonography. In addition, a cord-like, highly echogenic region penetrated the mass on B-mode ultrasonography, and a blood flow in the cord-like structure, which branched off inside the mass, was observed on color Doppler imaging (Figure 5). Pulsed-wave Doppler imaging showed a pulsatile and monophasic flow inside the cord-like structure, which identified the posterior branches of the hepatic artery and portal veins. In the CEUS images, relatively linear fine vessels in the arterial phase were seen evenly distributed in a web-like pattern within the tumor (Figure 6). In the portal phase, contrast enhancement was also seen in the central region of the mass with high echogenicity on B-mode ultrasonography.

The rapid growth of the mass, the low uptake in the HBP in EOB-MRI, and the lack of enhancement after the portal phase in CEUS led to the suspicion of a hepatic malignancy such as lymphoma, cholangiocellular carcinoma (CCC), cholangiolocellular carcinoma (CoCC), and neuroendocrine carcinoma. The patient underwent an extended posterior segmentectomy for histopathological examination, according to the patient’s wish.

The macroscopic examination of the resected specimen revealed a well-defined yellow-whitish mass without capsule formation (Figure 7). On hematoxylin and eosin (HE) staining, the medial part of the mass showed homogeneous and diffuse proliferation of medium- to slightly large-size atypical cells with irregular round shapes, some of which were degenerative and necrotic. Numerous mitoses were observed (>100 cells/10 high-power fields, HPFs). Diffuse proliferation of atypical cells that were one size larger than lymphocytes was seen in the HPFs. On immunohistochemical staining, CD20-expressing lymphocytes containing a large nucleus were noted. The following immunohistochemical staining was negative: AE1/AE3, CAM5.2, chromogranin A, synaptophysin, CD56, HSA, and CD3. Based on the pathological findings, the mass was finally diagnosed as a diffuse large B-cell lymphoma (DLBCL). The Ki-67 proliferative index was approximately 50–60%. After the surgery, fluoro-2-deoxyd-glucose positron emission tomography/computed tomography (FDG PET/CT) revealed no lymphomatous involvement apart from the liver. The patient underwent rituximab therapy for eight months. Subsequently, the DLBCL remained in remission for two years.

## 3. Literature Review

We conducted a thorough search of the PubMed database using the following three keyword combinations: (I) “primary hepatic lymphoma” AND ultrasonography, (II) “primary hepatic lymphoma” AND ultrasound, (III) “primary hepatic lymphoma” AND sonography. Our search included articles published until July 2023. Initially, the articles were filtered and selected based on their titles and abstracts, which was followed by a comprehensive evaluation of their full texts. We included only articles in the English language, focusing on ultrasonography imaging findings, while we excluded reviews, original articles, and articles without ultrasonography imaging findings.

Of a total of 45 papers initially identified, 21 were excluded for various reasons: 5 were review articles, 1 was an original article, 6 were non-English articles, and 9 lacked ultrasonography imaging findings. Consequently, we included 24 papers [12,13,15,16,17,18,19,20,21,22,23,24,25,26,27,28,29,30,31,32,33,34,35,36] in our analysis, which reported on 40 patients ranging in age from 22 to 86 years (Table 2). The recorded range of the largest lesion diameter in these cases was 1.5–17 cm.

The predominant lesion presentation was a solitary nodule, followed by multiple nodules and diffuse lesion (82.5%, 15%, and 2.5%, respectively). Solitary and multiple nodular lesions often show heterogeneous hypoechoic or anechoic ultrasonographic characteristics with posterior echo enhancement.

Among the included articles, three studies incorporated color Doppler imaging, while three used CEUS. The color Doppler imaging findings varied, ranging from “no obvious intratumoral blood flow” to “high perinodular and low intranodular vascularization”. Notably, Shiozawa et al. reported CEUS findings in PHL, indicating that “the lateral part of the lesion showed homogenous hyperenhancement, and the medial part of the lesion showed gradual staining from the margin toward the central region in the vascular phase [34]”. Additionally, He et al. utilized superb microvascular imaging and categorized their imaging findings into seven types, with PHL primarily reported as type VI, exhibiting a thick rim enhancement pattern [25].

Lu et al. documented 29 cases of PHL, in patients in an age range from 11 to 72 years [37]. Radiological examination showed three morphological patterns: solitary nodule, multiple nodules, diffuse lesion (*n* = 15, 9, 5, respectively). Solitary nodules ranged in size from 1.2 to 13.3 cm, whereas multiple nodules ranged from 1.0 to 8.4 cm.

Among the 24 cases with solitary and multiple nodular lesions, the majority exhibited homogeneous hypoechoic or isoechoic characteristics (20 vs. 4) and displayed well-defined, round, or oval masses (17 out of 24, 58.6%). Lu et al. further reported that B-cell lymphomas, encompassing DLBCL, mucosa-associated lymphoid tissue lymphoma (MALT), and follicular lymphoma, consistently manifested as nodular. In contrast, T-cell lymphomas, including NK/T lymphoma, peripheral T-cell lymphoma, and hepatosplenic lymphoma, exclusively presented as diffuse (*p* < 0.001). Regarding the pathology and mean lesion diameter, no significant differences were observed between well-defined lesions and those with a patchy distribution (*p* = 0.214 and 0.206, respectively).

## 4. Discussion

Malignant lymphomas, originating from the lymph reticulum system, often occur as extranodal primary lymphomas in various organs throughout the body. PHL is relatively rare because a normal liver generally contains a small interstitial component, and lymphocytes are only scattered in the portal area [38]. Consequently, liver lesions identified as malignant lymphoma are commonly seen as secondary extranodal infiltrations [39].

The occurrence of PHL was reported in only 6 cases (0.41%) in a study of 1467 cases of extranodal malignant lymphoma by Freeman et al. [1]. However, recent reports suggest an increasing incidence of PHL, possibly linked to the rise in immunocompromised conditions after organ transplantation and human immunodeficiency virus (HIV) infection.

In our case, there was no history of hepatitis virus or HIV infection or organ transplantation, and the etiology of PHL was unclear. In the majority of cases, lymphomas are diffuse large B-cell lymphomas, as in our case, with less than 10% being MALT lymphomas [1,40,41].

Caccamo et al. proposed criteria for defining PHL, stating that a malignancy can be characterized as such if, during autopsy, it is solely detected in the liver, or if lymphadenopathy or splenomegaly are absent for at least six months following the clinical diagnosis of a liver-based malignant lymphoma via a biopsy. Additionally, there should be no abnormalities in abdominal and chest CT images and in bone marrow or peripheral blood images other than liver lesions [42]. Ohsawa et al., on the other hand, outlined other PHL criteria, emphasizing the absence of involvement of other organs or lymph nodes, the absence of splenomegaly, and no indications of leukemia or myelosuppression [4].

Regarding PHL prognosis, Lu et al. found that of the 24 patients diagnosed with nodular primary hepatic lymphoma (PHL), 9 individuals who underwent surgical intervention remained free of tumors following a median follow-up of 12 months (ranging from 7 to 15 months). Among the 15 patients treated with chemotherapy, 8 achieved a complete response, 3 exhibited a partial response and were alive after a median follow-up of 9 months (ranging from 4 to 13 months), while the remaining 4 patients were lost to follow-up [37]. Our case showed no lymphomatous lesions other than the that in the liver and a complete response after chemotherapy.

In ultrasonography, solitary PHL are usually hypoechogenic, and some cases may appear almost anechoic [9,26]. In other cases, a “target lesion” shows a highly echogenic center and hypoechogenic margins [34,43]. The hypoechoic texture is likely attributed to high cellularity and the absence of surrounding stromal tissue [44]. PHL lesions might display increased peripheral vascularity, potentially resembling hemangiomas in color Doppler imaging. Color Doppler imaging can also demonstrate intratumoral hepatic artery penetration [31,37]. PHL in the arterial phase on CEUS demonstrated variability across multiple studies, while the late phase is more valuable, as nearly all lesions exhibit wash-out [13,34,45]. Our case showed that the mass was gradually enhanced through the vascular phase and showed lack of enhancement in the late phase.

PHL typically appears as a mass with soft tissue attenuation in non-contrast CT images, less enhanced compared to the surrounding liver tissue in both arterial and delayed phases on CT. On MRI, this lymphoma typically exhibits hypointensity in T1-weighted images (T1WI) and hyperintensity in T2WI. These hypovascular lesions show a similar subtle enhancement on CT, often accompanied by peripheral rim enhancement. PHL lacks the ability to uptake EOB, which results in a lower EOB uptake compared to the surrounding liver parenchyma, similar to metastatic liver tumors and CCC.

Likewise in ultrasonography, in cases where the lesion is relatively sizable, both contrast-enhanced CT and MRI can reveal the vessel-penetrating sign. Differential diagnoses based on the hepatic vessel-penetrating sign include PHL, CCC, CoCC, diffuse hepatocellular carcinoma, etc. One of the specific imaging findings to diagnose PHL is a low ADC value [44]. In DWI, PHL lesions present high signals due to increased cellular proliferation and elevated nuclear-to-cytoplasm ratio. Colagrande et al. reported that an ADC cut-off value of 0.918 × 10^–3^ mm^2^/s had a sensitivity and specificity of 81.7% and 100%, respectively, in differentiating between PHL from other malignant lesions [46]. The ADC value for our case was lower than the cut-off value.

Trenker et al. studied 38 HPLs and analyzed enhancement patterns in the arterial phase [14]. The study reported that hyper-, iso-, and hypoechogenic enhancement represented 23.7%, 44.7, and 31.6% of the cases, respectively. Our initial CEUS showed hyperechogenic enhancement, while the second CEUS displayed isoechogenic enhancement. Contrast medium accumulation in the arterial phase in CEUS depends on the vascularity of the tumor and its degree of necrosis. Likewise, the enhancement patterns detected by CECT and EOB-MRI vary according to the above tumor characteristics. In comparing the initial CEUS results with the CECT ones and the second CEUS results with the EOB-MRI findings, we found a discrepancy in the contrast pattern in the arterial phase. Both CEUS examinations showed intratumoral enhancement, while CECT and EOB-MRI demonstrated hypovascularity. Contrast media for ultrasonography such as SonoVue or Definity are true blood pool agents that do not accumulate unspecifically in the interstitium and do not cause unwanted background signals. Sonazoid, a liver-specific contrast agent, accumulates in the liver and is taken up by Kupffer cells. In the vascular phase, like other ultrasonographic contrast agents mentioned earlier, Sonazoid passes through the blood vessels. Otherwise, extracellular contrast agent can accumulate in the interstitium. These characteristics of the contrast media could lead to discrepancies between CEUS and CECT/EOB-MRI enhancement patterns in the arterial phase.

## 5. Conclusions

We reported a case of PHL in a normal liver that presented with specific imaging findings on multimodality imaging. Key features included wash-out in the late phase of contrast-enhanced imaging, absence of EOB uptake in the HBP, the vessel-penetrating sign, and a lower ADC value. Furthermore, a discrepancy between the CEUS and the CECT or MRI findings in the arterial phase could be indicative of PHL.

## Figures and Tables

**Figure 1 diagnostics-14-00306-f001:**
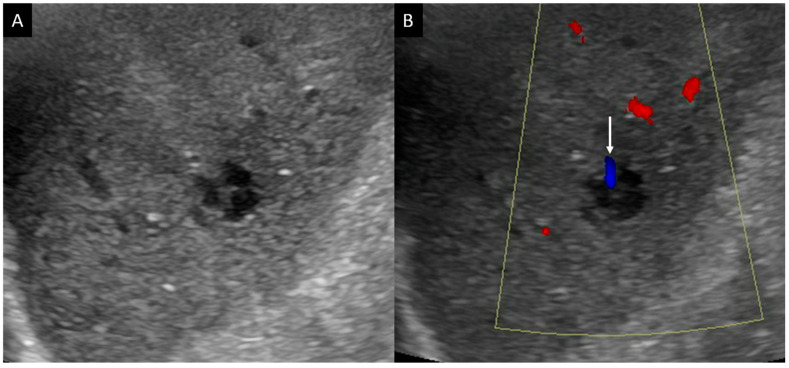
B-mode ultrasonography and color Doppler imaging. (**A**). B-mode ultrasonography shows anechoic and low-echogenicity features similar to those of the vessels. (**B**). Color Doppler imaging demonstrates blood flow within the nodule (arrow).

**Figure 2 diagnostics-14-00306-f002:**
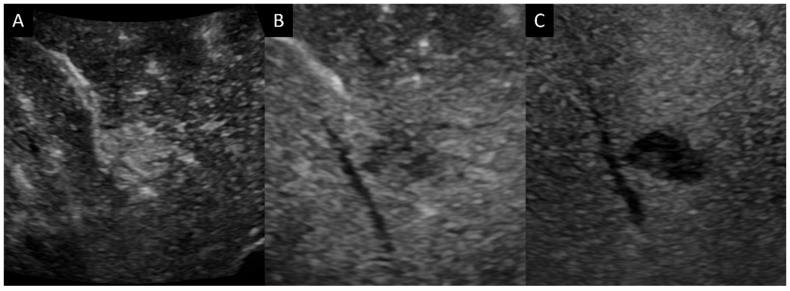
Contrast-enhanced ultrasonography (**A**). Arterial phase shows wash-in. (**B**,**C**). Enhancement within the nodule gradually fades, and wash-out is observed in the portal and late phase.

**Figure 3 diagnostics-14-00306-f003:**
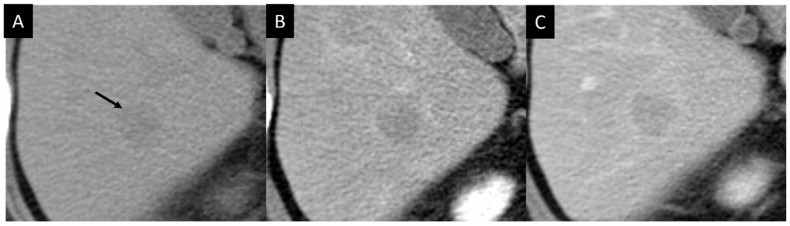
Contrast-enhanced CT. (**A**). Plain CT shows a nodule with slight lower attenuation than the surrounding liver parenchyma (arrow). (**B**,**C**). Arterial and portal phase images depict a hypovascular nodule.

**Figure 4 diagnostics-14-00306-f004:**
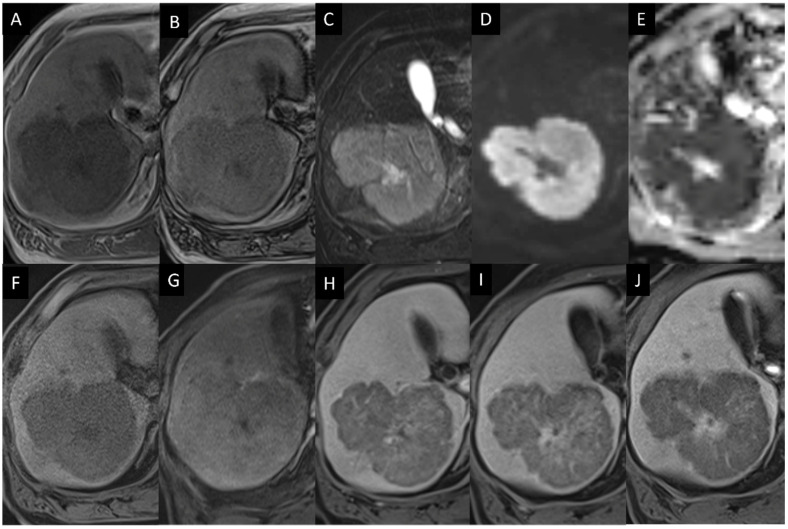
Gd-EOB-DTPA-enhanced MRI. (**A**,**B**). In-phase and opposed-phase T1-weighted images show a mass with lower intensity than the surrounding liver parenchyma. (**C**). Fat-saturated T2-weighted image shows high intensity. (**D**,**E**). Diffusion-weighted image shows high intensity, and apparent diffusion coefficient map shows a low value (0.5 × 10^−3^ mm^2^/s). (**F**–**I**). Dynamic contrast-enhanced images; pre-contrast, arterial, portal, and equilibrium phases show a hypovascular mass. (**J**). Hepatobiliary phase demonstrates a low EOB uptake within the mass.

**Figure 5 diagnostics-14-00306-f005:**
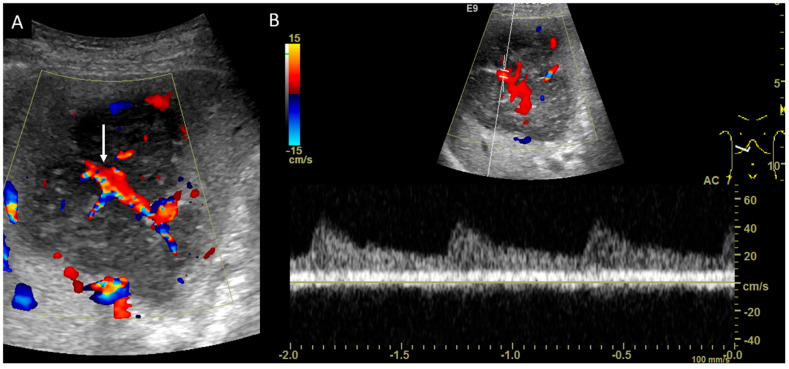
Color and pulsed-wave Doppler imaging. (**A**,**B**). Color Doppler image shows the blood flow (arrow), and pulsed-wave Doppler image shows a pulsatile flow inside the cord-like structure, which indicates the hepatic artery within the mass.

**Figure 6 diagnostics-14-00306-f006:**
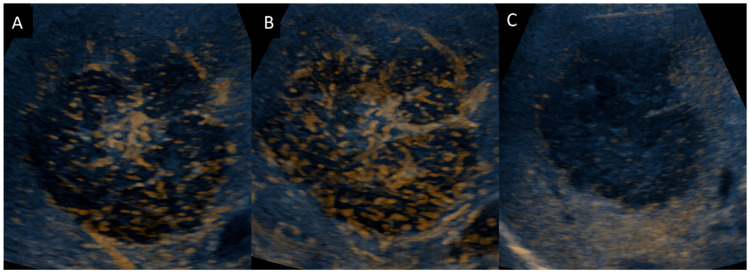
Contrast-enhanced ultrasonography. (**A**,**B**). Arterial and portal phases show a gradual enhancement. (**C**). Late phase demonstrates a lack of enhancement in the mass.

**Figure 7 diagnostics-14-00306-f007:**
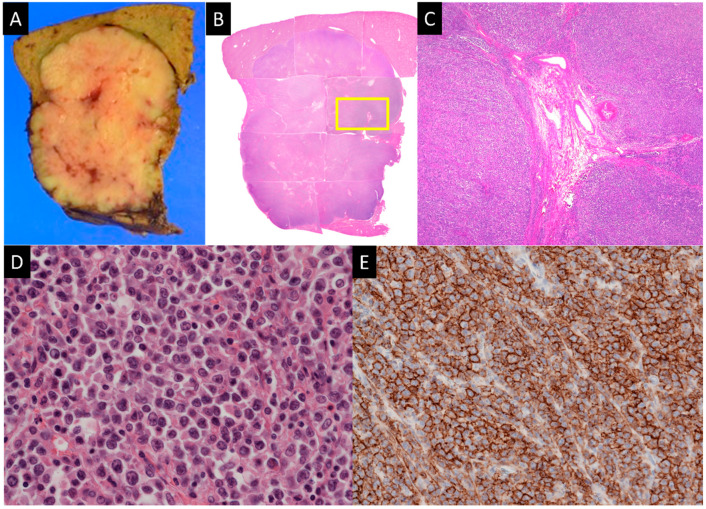
Pathology. (**A**). Macroscopic image revealing a well-defined yellow-whitish mass without capsule formation. (**B**–**D**). Loupe image, low-, and high-power field images, HE staining. Low-power field image is an image of the magnified yellow rectangle in the loupe image. The mass consists of medium- to slightly large-size atypical cells displaying homogeneous and diffuse proliferation and shows hepatic triads including the hepatic artery (square). (**E**). The cells are also positive for CD20 staining.

**Table 1 diagnostics-14-00306-t001:** Laboratory findings.

Complete Blood Count	Patient Value	Reference Range
White blood cell count (×10^3^/μL)	7.7	4–10
Hemoglobin (g/dL)	15.9	13–17
Platelet count (×10^3^/μL)	178	150–400
Blood biochemical test		
AST (U/L)	20	5–30
ALT (U/L)	18	5–30
Alkaline phosphatase (U/L)	290	50–100
γ-GT (U/L)	18	6–50
Total bilirubin (mg/dL)	0.7	2–20
Glucose (mg/dL)	143	70–100
Hemoglobin A1c (HbA1c, NGSP, %)	7.3	<5.5
Viral marker		
HBs Ag	(-)	
Anti-HBc	(-)	
Anti-HCV	(-)	
Tumor marker		
Alpha-fetoprotein (AFP, ng/mL)	3.5	0–44
Des-gamma-carboxy prothrombin (DCP, mAU/mL)	34	<40
Carcinoembryonic antigen (CEA, ng/mL)	2.2	<4
Cancer antigen 19–9 (U/mL)	12	<40
Soluble interleukin-2 receptor (sIL-2R, U/mL)	599	122–496

**Table 2 diagnostics-14-00306-t002:** Comparison of the data from a literature review.

Reference #	Case Report	Age	Gender	Phenotype	Maximum Size (cm)	B-Mode	Color Doppler US	CEUS
[12]	Low et al., 2006	67	M	Solitary	N/A	Hypoechoic	Vessel displacement	Homogeneous enhancement followed by enhancement
[13]	Foschi et al., 2010	58, 62	2 males	Solitary, solitary	3, 4	2 hypoechoic	N/A	2 Inhomogeneous enhancement
[15]	Dhamija et al., 2015	22	F	Solitary	11	Heteroechoic	N/A	N/A
[16]	Nishikawa et al., 2021	85	M	Nodules	N/A	Hypoechoic	N/A	N/A
[17]	Scucchi et al., 2020	86	M	Solitary	5	Heteroechoic	N/A	N/A
[18]	Sato et al., 1999	41	F	Solitary	1.5	Hypoechoic	N/A	N/A
[19]	Zafar et al., 2012	68	F	Solitary	17	Hypoechoic	N/A	N/A
[20]	Iannitto et al., 2004	62	M	Solitary	13	Hypoechoic	N/A	N/A
[21]	Soyer et al., 1993	27–78 (*n* = 4)	N/A	4 Solitary cases	5–10	4 Hypoechoic	N/A	N/A
[22]	Tajiri et al., 2014	75	M	Solitary	2.5	Hypoechoic	N/A	N/A
[23]	Tsuruta et al., 2002	62	F	Solitary	5	Hypoechoic	N/A	N/A
[24]	Mahler et al., 2001	29, 34, 42, 49, 61, 61, 71	7 males	4 Solitary cases, 3 nodules	N/A	6 Hypoechoic and anechoic	N/A	N/A
[25]	He et al., 2017	71	M	Solitary	N/A	Hypoechoic	N/A	Thick rim enhancement in the arterial phase
[26]	Appelbaum et al., 2005	23–59 (*n* = 7)	6 males and 1 female	7 solitary cases	N/A	5 Hypoechoic and 2 anechoic	N/A	N/A
[27]	Park et al., 2019	73	F	Solitary	3.3	Hypoechoic	N/A	N/A
[28]	Dantas et al., 2020	65	M	Solitary	7	Hypoechoic	N/A	N/A
[29]	Ozaki et al., 2020	73	F	Solitary	4.5	Hypoechoic	No obvious intratumoral blood flow	N/A
[30]	Liao et al., 2017	73	M	Nodules	2	Hypoechoic	N/A	N/A
[31]	Raimondo et al., 2012	52	F	Solitary	2.8	Hypoechoic	High perinodular and low intranodular vascularization	N/A
[32]	Diehl et al., 2013	64	F	Solitary	N/A	Hypoechoic	N/A	N/A
[33]	Forghani et al., 2017	67	M	Solitary	1.9	Hypoechoic	N/A	N/A
[34]	Shiozawa et al., 2015	60	F	Solitary	1.5	Heteroechoic	N/A	Gradually enhanced from the margin toward the central region
[35]	Lisker-Melman et al., 1989	41	M	Nodules	3	Hypoechoic	N/A	N/A
[36]	Das et al., 1993	52	M	Diffuse	N/A	Heteroechoic	N/A	N/A

N/A; not available.

## Data Availability

The original data presented in the study are included in the article. Further inquiries can be directed to rtaiji@naramed-u.ac.jp.

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
