# Peer review of "Multimodality Imaging of Primary Hepatic Lymphoma: A Case Report and a Literature Review"

_diagnostics, 2024, doi:10.3390/diagnostics14030306_

Round 1

Reviewer 1 Report

Comments and Suggestions for Authors

General remarks

This is a well-written and informative article on the imaging of a primary lymphoma of liver. I hope the readers will be benefitted by the paper.

Author Response

Dear reviewer,

We are truly grateful for the time, effort, and expertise they dedicated to evaluating our work. We appreciate your indispensable contributions to the success of our research.

Ryosuke Taiji

Reviewer 2 Report

Comments and Suggestions for Authors

Manuscript Diagnostics-2828271

The paper reports on a primary hepatic lymphoma case (PHL) in a normal liver of an 84-year-old man whose pathological diagnosis of PHL was confirmed postoperatively through pursuant radiological examinations. The manuscript underscores contrast-enhanced ultrasonographic findings of PHL such as enhancement patterns and vessel penetrating signs from an early phase. The manuscript delves into other studies of this tumor trying to address its accompanying clinical, pathological, and radiological features.

Few points to consider in a possible revision of this manuscript:

·    It is not clear why there is only one single case (that of an 84-year-old man; age-/gender-specific) addressed in this study. This is especially curious since a plethora of other studies were also reviewed by the authors. As such, it is hard to judge the comprehensiveness of the discussed imaging patterns and whether such exact same findings will be applicable or expected in other age/gender cases. This is important since the authors state that their single-case study is perhaps the first report detailing enhancement patterns specifically discussed in their manuscript.

·      Data presentation and comparison of various findings of the reviewed literature (Sections 3 and 4) may be more clearly done by employing table-like structures.

·   The conclusion section is weak and needs to be extended in a more structured way, employing robust argumentations that would underscore the uniqueness of this work in its own right.

Comments on the Quality of English Language

No major issues detected.

Author Response

Reviewer 2

We deeply appreciate your time, effort, and expertise they have generously shared in ensuring the scholarly excellence of our publication.

Ryosuke Taiji

Reviewer 3 Report

Comments and Suggestions for Authors

Case Report. Multimodality Imaging of Primary Hepatic Lymphoma: a Case Report and a Literature Review. Diagnostics 2024.

The report highlights specific ultrasonographic signs of PHL observed from an early stage, accompanied by a review of relevant literature.

Major poin

Please explain more specifically “what is new” in the case report? What can we learn and which has not been described so far (especially in the light that the Trenker study has been cited)?

Minor points

Eliminate “How to Use This Template”

Ultrasonography serves as the first examination for screening of focal liver lesions [please give reference].

Color Doppler imaging assesses the vascular abundance in the mass, and pulsed wave [please give reference]

Doppler imaging identifies the type of vessels within the hepatic artery and portal vein [please give reference]

Unlike extracellular contrast agents used in contrast-enhanced CT (CECT) or MRI, ultrasound contrast agents circulate solely within blood vessels, allowing more reliable view of intratumoral vascular structures [please give reference]. While CT and MRI only detect penetrating vessel signs in larger masses, CEUS can identify the tiny vessels even in a smaller masses [please give reference].

I am not familiar with “biliary deviation enzymes”, better: “cholestasis indicating enzymes”.

Please explain why you examined tumor marker, especially why “des-gamma-carboxy prothrombin (DCP, mAU/mL)” and “soluble interleukin-2 receptor (sIL-2R, U/mL)”? Please refer to guidelines for recommendation to do so.

As a matter of curiosity: Please explain “posterior segmental arteries”

As a matter of curiosity: Please explain cholangiocellular carcinoma (CCC), cholangiolocellular carcinoma (CoCC) and refer to commonly used terminology. It might be my ignorance not to use this differentiation. 

Figure 4: Different types of letters are shown, please correct. 

Abstract: Subsequent postoperative radiological examinations, without any additional therapies, In the text rituximab is mentioned. One of the statements is wrong. 

“posterior echo enhancement“: Please check terminology, if you scan from posterior it would be anterior echo enhancement?

Discussion

Malignant lymphomas, originating from the lymph reticulum system, and often occur as extranodal primary lymphomas in various organs throughout the body. However, PHL is relatively rare because a normal liver generally contains little interstitial component and lymphocytes are only scattered in the portal area [please give reference]. Consequently, liver lesions attributed to malignant lymphoma are commonly seen as secondary extranodal infiltration [please give reference].

“PHL lesion might display increased peripheral vascularity, potentially resembling hemangiomas in color Doppler imaging” I am not aware that hemangioma typically display increased peripheral vascularity detectable by Doppler US. I am aware that there are feeding and draining vessels [please give reference].

The sentence “PHL lesions could contain hemorrhagic or necrotic elements” is lost between imaging describing sentences. Imaging can describe non-enhancing lesions but does not allow to differentiate necrotic elements?

ADC is explained twice.

Is Sonazoid a real blood pool agent? Please explain in this regard the Kupffer Phase of Sonazoid. Please check this point to the discussion you made. 

Author Response

Dear Reviewer 3

We extend our sincere gratitude to your constructive feedback have significantly contributed to the enhancement of the quality and rigor of our manuscript. We deeply appreciate the time, effort, and expertise they have generously shared in ensuring the scholarly excellence of our publication.

Comments and Suggestions for Authors

Case Report. Multimodality Imaging of Primary Hepatic Lymphoma: a Case Report and a Literature Review. Diagnostics 2024.

The report highlights specific ultrasonographic signs of PHL observed from an early stage, accompanied by a review of relevant literature.

Major points

Please explain more specifically “what is new” in the case report? What can we learn and which has not been described so far (especially in the light that the Trenker study has been cited)?

Authors: This case report stands out due to its novelty in capturing changes in PHL over time through multimodality imaging. We consider this an educational case as it presents the specific characteristic imaging findings reported previously. Notably, the distinctive feature is the discrepancy in contrast patterns during the early phase, with the tumor exhibiting early hyperechogenic enhancement on CEUS from an early stage, in contrast to hypovascular tumor observed on CT and MRI.

We added in the conclusion as, “We reported a case of PHL in a normal liver that presented with specific imaging findings on multimodality imaging. Key features included wash-out in the late phase of contrast-enhanced imaging, absence of EOB uptake in HBP, a vessel-penetrating sign, and a lower ADC value. Furthermore, a discrepancy between CEUS and CECT or MRI in the arterial phase could be indicative of PHL.”

Minor points

Eliminate “How to Use This Template”

Authors: This phrase was eliminated from the resubmitted version.

Ultrasonography serves as the first examination for screening of focal liver lesions [please give reference].

Authors: we added the reference by Harvey et al.

Color Doppler imaging assesses the vascular abundance in the mass, and pulsed wave [please give reference]

Authors: we added the reference by Nino-Murcia et al.

Doppler imaging identifies the type of vessels within the hepatic artery and portal vein [please give reference]

Authors: we added the reference by Owen et al.

Unlike extracellular contrast agents used in contrast-enhanced CT (CECT) or MRI, ultrasound contrast agents circulate solely within blood vessels, allowing more reliable view of intratumoral vascular structures [please give reference]. While CT and MRI only detect penetrating vessel signs in larger masses, CEUS can identify the tiny vessels even in a smaller masses [please give reference].

Authors: we added the reference by Paefgen et al. for the first sentence. Regarding the second sentence, we would like to modify as, “In breast cancer, CEUS demonstrates higher sensitivity than MRI for evaluating microvessel density, including identifying penetrating vessel signs.” and add the reference by Jia et al.

I am not familiar with “biliary deviation enzymes”, better: “cholestasis indicating enzymes”.

Authors: we modified as you pointed out.

Please explain why you examined tumor marker, especially why “des-gamma-carboxy prothrombin (DCP, mAU/mL)” and “soluble interleukin-2 receptor (sIL-2R, U/mL)”? Please refer to guidelines for recommendation to do so.

Authors: Tumor markers were measured to conduct a comprehensive examination of the liver mass. Despite blood and viral tests indicating that the patient did not fall into the high-risk group for HCC, early enhancement during the arterial phase on CEUS was observed. Consequently, both HCC and hepatic lymphoma were considered in the differential diagnosis, and measurements for these two tumors were also taken. We cannot find any guidelines, but we found a case report showing a high level of sIL-2R with primary hepatic lymphoma.

Primary Hepatic Peripheral T-Cell Lymphoma Treated with Corticosteroid

Miyashita et al.

https://doi.org/10.2169/internalmedicine.50.4686

As a matter of curiosity: Please explain “posterior segmental arteries”

Authors: we modified as, “Pulsed wave Doppler imaging showed pulsatile and monophasic flow inside the cord-like structures, which indicated posterior branches of hepatic artery and portal veins.”

As a matter of curiosity: Please explain cholangiocellular carcinoma (CCC), cholangiolocellular carcinoma (CoCC) and refer to commonly used terminology. It might be my ignorance not to use this differentiation.

Authors: Cholangiocellular carcinoma (CCC), or intrahepatic cholangiocarcinoma, originates from the liver's bile ducts, specifically the intrahepatic ones. It's an adenocarcinoma arising from epithelial cells, diagnosed through histological and immunohistochemical features, often associated with a poor prognosis.

In contrast, cholangiolocellular carcinoma (CoCC) is a rare primary liver neoplasm originating from hepatic progenitor cells or cholangioles. Recognizable by distinct histological features like small ductules resembling canals of Hering, CoCC exhibits a unique immunohistochemical profile compared to CCC. Despite its rarity, CoCC has a more favorable prognosis than CCC, making its accurate preoperative diagnosis challenging.

Reference:

Intrahepatic cholangiocarcinoma and cholangiolocellular carcinoma in cirrhosis and chronic viral hepatitis

Ariizumi et al.

https://link.springer.com/article/10.1007/s00595-014-1031-0

Figure 4: Different types of letters are shown, please correct.

Authors: we corrected that.

Abstract: Subsequent postoperative radiological examinations, without any additional therapies, In the text rituximab is mentioned. One of the statements is wrong.

Authors: In the abstract, we would like to emphasize that the patient presented no other lesions at the time of diagnosis only with hepatic segmentectomy. However, the explanation in the abstract was confusing, so we deleted “without any additional therapies.”

“posterior echo enhancement“: Please check terminology, if you scan from posterior it would be anterior echo enhancement?

Authors: some articles mentioned the phenomenon that the increased echoes deep to structures that transmit sound exceptionally well on B-mode, as “posterior enhancement”, “posterior echo enhancement” or “posterior acoustic enhancement.”

References:

Clinical ultrasound physics

Abu-Zaiden et al.

https://www.ncbi.nlm.nih.gov/pmc/articles/PMC3214508/

Ultrasonographic differentiation of hepatocellular carcinoma from metastatic liver cancer

Yoshida et al.

https://onlinelibrary.wiley.com/doi/10.1002/jcu.1870150702

Posterior Acoustic Enhancement in Hepatocellular Carcinoma

Maturen et al.

https://onlinelibrary.wiley.com/doi/abs/10.7863/jum.2011.30.4.495

Discussion

Malignant lymphomas, originating from the lymph reticulum system, and often occur as extranodal primary lymphomas in various organs throughout the body. However, PHL is relatively rare because a normal liver generally contains little interstitial component and lymphocytes are only scattered in the portal area [please give reference]. Consequently, liver lesions attributed to malignant lymphoma are commonly seen as secondary extranodal infiltration [please give reference].

Authors: we referred the chapter by Du et al. for the former sentence and by Edmondson et al for the latter.

“PHL lesion might display increased peripheral vascularity, potentially resembling hemangiomas in color Doppler imaging” I am not aware that hemangioma typically display increased peripheral vascularity detectable by Doppler US. I am aware that there are feeding and draining vessels [please give reference].

Authors: I agree your comment partially but color Doppler imaging can depict intratumoral vascularity and peripheral feeding vessels as previous articles reported:

References:

Lim et al.

Colour Doppler sonography of hepatic haemangiomas with arterioportal shunts

10.1259/bjr/96605786

Nino-Murcia et al.

Color Doppler flow imaging of focal hepatic lesions.

10.2214/ajr.159.6.1332456

Perkins et al.

Color and power Doppler sonography of liver hemangiomas: a dream unfulfilled?

10.1002/(sici)1097-0096(200005)28:4<159::aid-jcu1>3.0.co;2-b

The sentence “PHL lesions could contain hemorrhagic or necrotic elements” is lost between imaging describing sentences. Imaging can describe non-enhancing lesions but does not allow to differentiate necrotic elements?

Authors: Imaging can detect hemorrhagic area but cannot accurately determine the necrotic elements only showing lower-enhanced region. Additionally, we should not mix up the radiological and pathological findings in the paragraph mentioning imaging findings. So, we deleted the sentence, “PHL lesions could contain hemorrhagic or necrotic elements.”

ADC is explained twice.

Authors: we deleted the duplicate second explanation.

Is Sonazoid a real blood pool agent? Please explain in this regard the Kupffer Phase of Sonazoid. Please check this point to the discussion you made.

Authors: No, Sonazoid is not a true blood pool agent. SonoVue is. Sonazoid passes through vessels in vascular phase and accumulates in the liver and is taken up Kupffer cells.

We modified and added as, “Contrast media of ultrasonography such as SonoVue or Definity play role as true blood pool agents, which is prevented unspecific accumulation in the interstitium and unwanted background signals. Sonazoid, a liver-specific contrast agent, accumulates in the liver and is taken up by Kupffer cells. In the vascular phase, like other ultrasonographic contrast agents mentioned earlier, Sonazoid also passes through blood vessels.”

Ryosuke Taiji

Round 2

Reviewer 2 Report

Comments and Suggestions for Authors

The authors have addressed the suggested clarifications satisfactorily. Only a minor point that the newly added Table 2 needs also to be referenced and re-positioned within the text of the manuscript, like in the case of Table 1 which is referenced in line 62 of the manuscript.

Reviewer 3 Report

Comments and Suggestions for Authors

Also I do not personally consent all statements in the revision and responses to my review the authors cited literature. Therefore, thank you for the revision.